# Experimental evaluation of DPF performance loaded over Pt and sulfur-resisting material for marine diesel engines

**Xiaobo Li[1,2]ʘ, Ke Li[2]ʘ, Haoran Yang**  **[2]ʘ, Zhigang Wang[2]ʘ, Yaqiong Liu[2]‡, Teng Shen[3]‡, Shien Tu[3]‡, Diming Lou[1] ***

**1** School of Automotive Studies, Tongji University, Shanghai, P.R. China, **2** Research and Design Center, Shanghai Marine Diesel Engine Research Institute, Shanghai, P.R. China, **3** National Engineering Laboratory for Marine and Ocean Engineering Power System, Shanghai Marine Diesel Engine Research Institute, Shanghai, P.R. China

ʘ These authors contributed equally to this work.
‡ These authors also contributed equally to this work.
* loudiming@tongji.edu.cn

**Data Availability Statement:** All relevant data are within the paper and its Supporting Information files.

**Funding:** The author(s) received no specific funding for this work.

## Abstract

Different from vehicle engines, Diesel Particulate Filter (DPF) inactivation is an unavoidable issue for low-speed marine diesel engines fueled with Heavy Fuel Oil (HFO). This paper introduced a sulfur resisting material in Silicon Carbide (SiC)-DPF to improve DPF performance. The results of bench-scale experiments showed that the Balance Point Temperature of the modified DPF module was 300°C and DPF modules had a good filtration performance, with Particulate Matters (PMs) residual being less than 0.6 g per cycle. In pilot-scale tests, PMs emissions of unit power decreased with engine load going up, filtration efficiency of nucleation mode PMs being only 36% under 100% load, while DPF still had a good performance in accumulation mode PMs control, being 94.2% under the same load. DPF modules showed excellent regeneration durability in the 205h endurance test, with a regeneration period of 1.5-2h under 380°C. There was no obvious degeneration in the DPF module structure, with no cracks or breakage. Besides, the DPF module could also control gaseous emissions, total emissions decreased by 10.53% for NO and 57.19% for CO, respectively. The results suggested that introducing sulfur-resisting material in DPF could greatly improve the DPF performance of low-speed marine diesel engines fueled with HFO.

## Introduction

Nowadays, human beings consume more energy to maintain normal daily life than ever, for the sharply increasing population, industry, and transportation. Energy usually comes from fossil fuel combustion [1]. But fossil fuel combustion brings many side effects, especially environmental issues [2]. Particulate Matter (PM) emission, produced by diesel engines, is one of the trickiest issues that human beings have to deal with. PM is a general term for various particulates in the atmosphere. It can remain in the atmosphere for a long time and cause irreversible harm to the global environment and human health [3].

**Competing interests:** The authors have declared that no competing interests exist.

Many laws and regulations have been carried out to control PM emissions in the fields like motor vehicles and fixed sources in recent years [4]. As a result, the proportion of marine PM emissions is gradually increasing, especially in the port area, which has aroused the concern of global society. Although there is no global regulation, many countries and organizations are drafting local regulations on limiting marine PM emissions. International Maritime Organization (IMO) is introducing Regulation 14 inside Emission Control Areas (ECA) to limit SOx and PMs emission. China has carried out "Limits and measurement methods for exhaust pollutants from marine engines (STAGE II, effective since July 2021)" for emission control in the coastal area [5–7].

PM of marine vessels and engines mainly comes from incomplete combustion of diesel fuels [8]. Mass PM emission leads to severe pollution and affects engine efficiency. DPF has been confirmed as an effective PM-reducing method for diesel engines [9–11]. The DPF equipment can collect the particulates and then remove those particulates by oxidation continuously or discontinuously during engine working. After the collected PM particulates are removed, DPF can collect the following PMs in the exhaust gas. However, the oxidation temperature of PM particulates is usually above 600°C, much higher than that of exhaust gas [12,13], Many catalysts, like precious metals (Pt, Rh, and Pd). are introduced to realize low-temperature oxidation of PMs in the DPF working process [14–16]. Those precious metals help to catalyze NO to $NO_2$ under low temperatures, and $NO_2$ will further oxidize PM particulates to $CO_2$ [17].

Although the effect of DPF equipment has been tested in motor diesel engines and the fixed resource field [18], there is few successful DPF application case in the low-speed marine diesel engines field [19]. Many studies find that high sulfur content in HFO is the main reason for that failure [20,21]. Statistics indicate that the S content in fuel oil is 0.1% or above for most oceangoing ships. Combustion of S content will produce gaseous sulfur oxides. Those sulfur oxides will further react with DPF catalysts to produce sulfate [22]. The mentioned chemical process can cause metal oxides inactivation and destroy the microcrystal structure of precious metals, even leading to permanent damage to DPF catalysts. Therefore, it is urgent to develop a DPF catalyst suitable for marine diesel engines with HFO [23,24].

Therefore, this paper introduced a sulfate resisting material to a commercially available DPF module (SiC) to improve DPF performance under the HFO conditions of marine diesel engines. We carried out two-stage experiments to have a comprehensive understanding of DPF performance, a simulated-exhaust-gas bench scale, and a pilot scale, respectively. We investigated the effects of Pt and sulfate resisting material content on DPF performance in two-stage experiments, including balance point temperature (BPT), accumulated carbon content, and regeneration characteristics [25]. These findings would provide a theoretical and technical basis for the practical application of DPF loaded over sulfur resisting material.

## Materials and methods

### Sample preparation

Fig 1 shows the DPF modules used in this paper. Specifications for DPF preparation are described as follows:

Step 1: Preparing the active ingredient precursor solution. We weighted certain Pt solutions according to PGM loading, and added certain deionized water to the weighted Pt solution;

Step 2: Preparing sulfate resisting material SR-1. Sulfate resisting material SR-1 was a mixture of multiple industrial-batch raw materials, containing $Al_2O_3$, $BaSO_4$, and organic colloid. Like Step 1, we diluted the SR-1 solution with deionized water and put it into the prepared

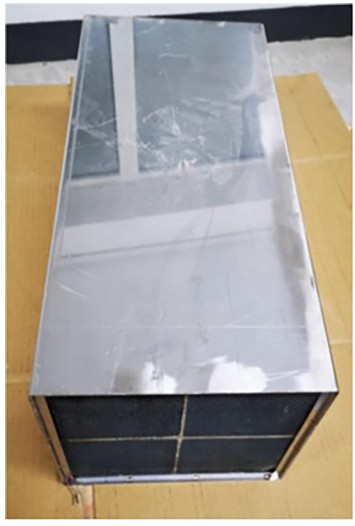

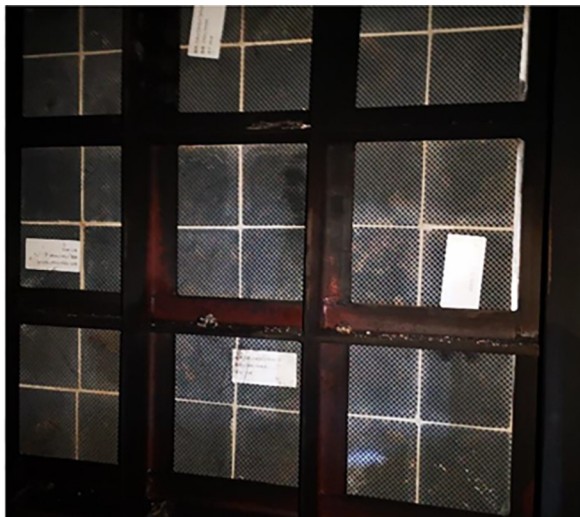

(A)

(B)

**Fig 1. DPF catalyst module.** (A) individual catalyst, (B) Packaged catalyst.

active ingredient precursor solution. And then, we stirred the mixed solution for 30 minutes to make sure that the solution was well mixed;

Step 3: Coating carries material with the mixed solution. We chose SiC as the DPF carries (pore density: 200 mesh). We kept SiC material immersed in the mixed solution for a certain time until the air in the pore was drained. After the mixed solution was fully and uniformly distributed in the SiC material surface and pores, we took out the SiC material and blew away the excess solution with an air gun;

Step 4: The coated catalyst went through two-hour calcination in a Muffle furnace at 500˚C and cooled to room temperature. Finally, a rectangular catalyst with a volume of 2.9L was prepared, like Fig 1A.

In this paper, we adopted three schemes for coating and loading, as shown in Table 1 to analyze DPF performance with different coating materials and Pt loadings.

## Experiments and tests

This paper carried out experiments and tests on bench-scale and pilot-scale, respectively to comprehensively understand DFP performance.

**Table 1. Coating materials and Pt loadings of DPF module.**

| No. | Carriers | Coating | PGM |
|---|---|---|---|
| 1 | SiC | None | Pt 10g/cft |
| 2 | SiC | SR-1 | Pt 10g/cft |
| 3 | SiC | SR-1 | Pt 40g/cft |

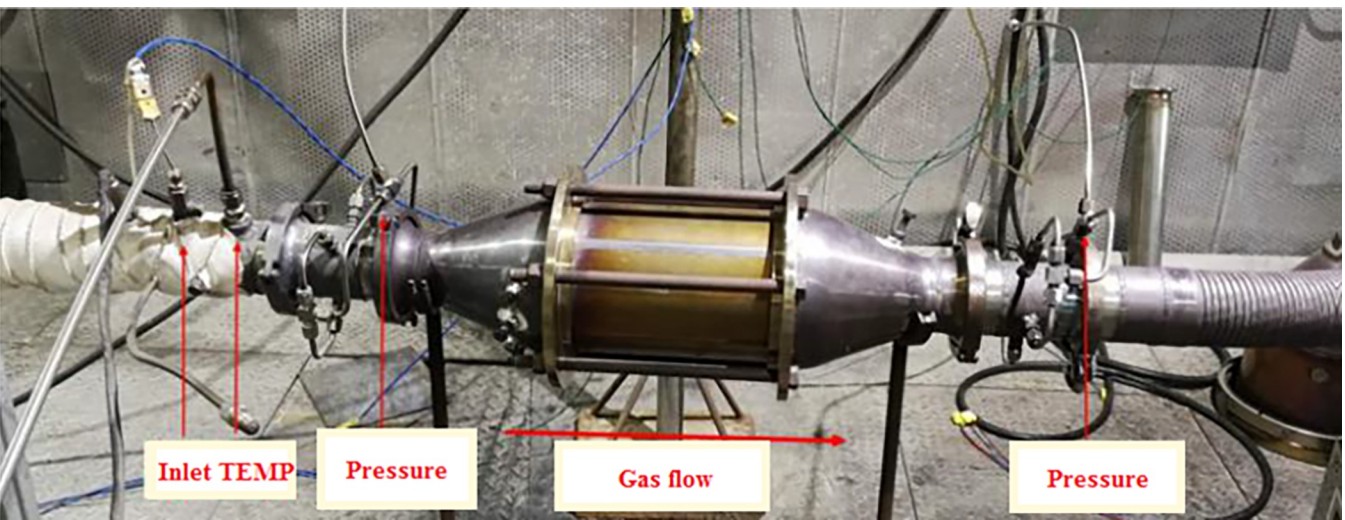

**Fig 2. The simulated-exhaust-gas experiments bench.**

The simulated-exhaust-gas bench-scale experiments were for BPT and performance analysis of DPF modules, which would lay the foundation for pilot-scale tests. The experiment setup consisted of a diesel engine(2.8L), temperature sensors, flow sensors, pressure sensors, and a high-precision electronic weighing instrument, as shown in Fig 2. The engine used is BJ493ZLQ4 from Beijing Futian Environmental Power Co., Ltd. It is a four-stroke, high-pressure common rail system, turbocharged and intercooled engine. The displacement is 2.8L, the rated power is 70 kW, and the declared speed is 3600rpm. The fuel used was an artificial high-sulfur diesel fuel, which was made by adding thiophene into National VI diesel fuel (GB/19147-2016), and the total sulfur content of diesel fuel is 0.5%.

The pilot-scale tests were for analyses of PM filtering efficiency, gaseous pollutant removal efficiency, regeneration BPT, and regeneration durability for DPF catalyst. The test setup consisted of a large diesel engine, exhaust gas duct, and exhaust gas analysis equipment, as shown in Fig 3. The engine used was WHM6160C550-5 from Weichai Holding Group Co., Ltd. It

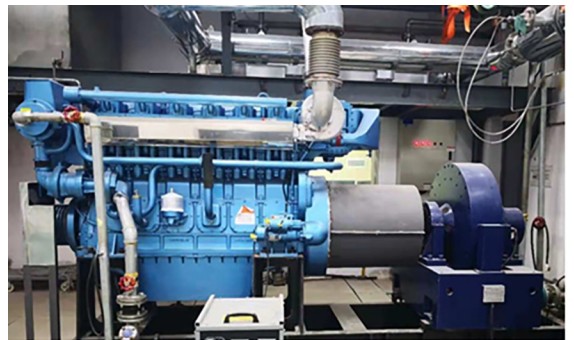

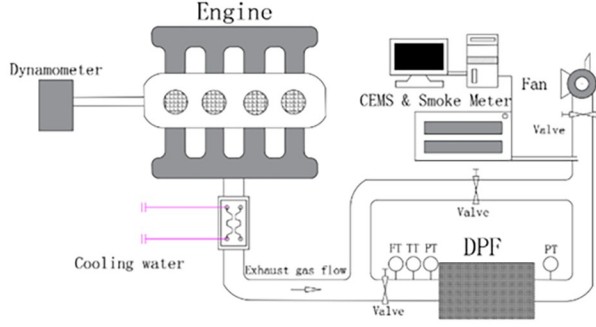

(A)

(B)

**Fig 3. The pilot-scale tests bench.** (A)Diesel engine, (B) Bench schematic figure.

was a four-stroke, water-cooled, in-line, turbocharged, and intercooled engine. The displacement was 24.12L, the rated power was 405 kW, and the declared speed was 1500rpm. The fuel used was the same as in bench-scale experiments. The exhaust gas duct was connected to the exhaust gas outlet of the diesel engine. DPF modules and an exhaust gas treatment unit were set in the middle and outlet of it after a water-cooling system. The duct was covered with insulation materials to minimize heat dissipation in tests. Gas measurement points were symmetrically set upstream and upstream of the DPF module, including CEMS, pressure transmitters, flow transmitters, temperature transmitters, and smoke meter. The CEMS used was the AVL AMA i60 and the smoke meter used was the Testo 338smoke tester. All experiments and test results were arithmetic mean of 3 measurements on the same condition in this paper, but regeneration cycle tests.

# Results and discussion

## Simulated-exhaust-gas bench-scale experiments

**DPF module BPT experiments.** As the DPF module filtered PMs, the engine exhaust backpressure, exhaust gas temperatures, and DPF module temperature would rise. Then, the trapped PMs would become oxidized with the environment temperature increase, which led to a decrease in engine exhaust backpressure. This process was called DPF regeneration. When the particle deposition and regeneration rates got balanced and the pressure drop between the upstream and downstream DPF module remained stable, the inlet temperature of the DPF module was defined as BPT in the regeneration process. Therefore, BPT was one of the most vital parameters for judging DPF module performance. The BPT results of three DPF modules were shown in Fig 4.

As shown in Fig 4, exhaust gas temperature went up with the increase in diesel engine power. For DPF 1#, the inlet temperature of the DPF module grew gradually and stably. From 200˚C to 320˚C, DPF pressure drops gradually increased from 5.4kPa to 6.0kPa, indicating that PMs were still in the deposition stage and DPF regeneration was slow in this zone. There was no apparent decrease in DPF pressure until the temperature rose to 325˚C. Therefore, 325˚C was the BPT of DPF 1#. When the temperature went over 325˚C, the DPF pressure drop decreased, showing that PMs oxidation rate exceeded the deposition rate and the DPF module went into the deposition stage. Similarly, although coated with sulfur resisting material, DPF 2# had the same BPT result as DPF 2# because of the same Pt coating content with 1#. However, compared with 1# and 2#, the BPT result of DPF 3# showed a significant drop,

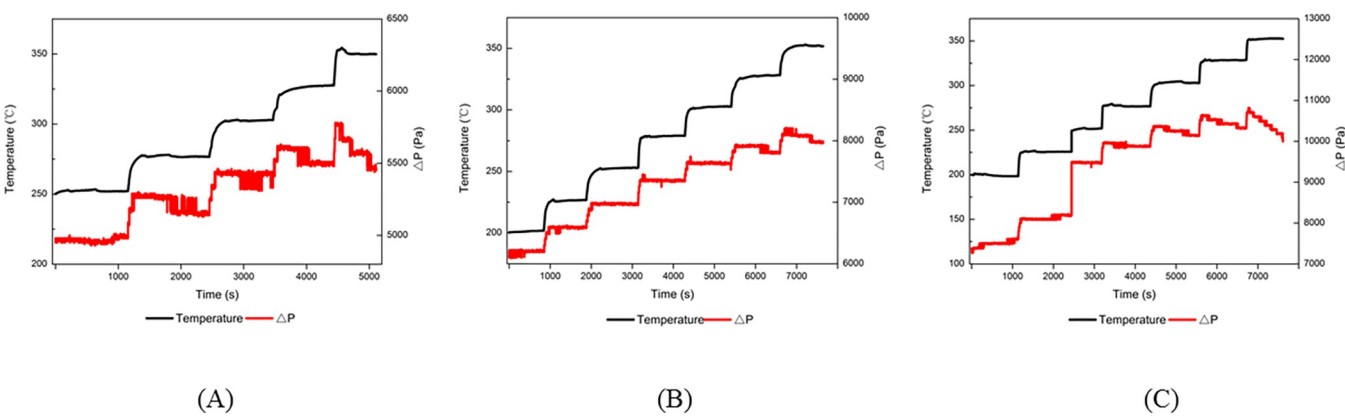

(A)                                    (B)                                    (C)

**Fig 4. BPT results for DPF with different coating and loading contents.** (A) DPF 1#, (B) DPF 2#, (C) DPF 3#.

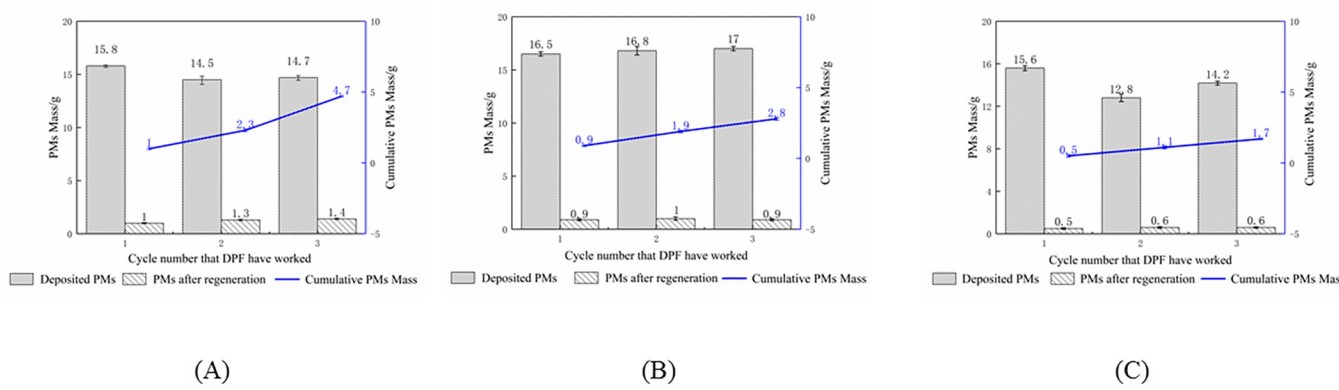

(A)                                    (B)                                    (C)

**Fig 5. PMs deposition and residuals in DPF with different coating and loading contents.** (A) DPF 1#, (B) DPF 2#, (C) DPF 3#.

decreasing to around 300˚C. Meanwhile, DPF pressure drop also showed a decreasing trend around 300˚C. The results show that Pt contents in the DPF module had a better effect on BPT than sulfur-resisting material materials.

**Sulfur resistance experiments for DPF.** Mass studies have shown that although DPF catalysts with high precious metals loading are more effective in PMs filtering, high sulfur contents in the exhaust gas will destroy the microcrystalline structure of the noble metals and lead to permanent DPF deactivation. So we introduced a new sulfur resisting material and modified DPF with this material in this paper. And we carried out three-times-cycle experiments on the sulfur-resisting performance of the DPF module.

Fig 5. described the results of DPF sulfur resisting performance. The PMs residual of DPF 1# increased with experiments cycles and kept above 1.0g per cycle. However, the average PMs residual of DPF 2# was less than 1.0g per cycle, without the upward trend as the number of cycles increased, which indicated that SR-1 had the effect of immobilizing the precious metal Pt and better sulfur-resisting performance. Compared with DPF 2#, the average PM residual of DPF 3# was significantly less for each cycle. That was saying that increasing the amount of precious metal Pt could considerably enhance the oxidation capacity of the DPF catalyst, and the DPF modules removed almost all PMs.

## Pilot-scale tests

In Section 3.1, we studied DPF performance in simulated-exhaust-gas bench-scale experiments and the results showed that SR material and precious metals could improve the DPF catalyst performance. However, bench-scale experiments could hardly have a comprehensive understanding of DPF performance, so in this section, we carried out tests on DPF 3# in a pilot-scale bench to have further research.

**DPF module PMs filtering efficiency.** To evaluate the PMs filtering efficiency of the DPF module, we tested DPF upstream and downstream PMs emission by weighing the filter papers, at 25%, 50%, 75%, and 100% load conditions, respectively. Besides, we conducted a 65% load condition test to reduce the influence that 75% load condition might have on analysis since the exhaust gas temperature under 65% load was close to DPF balance point temperature and 75% load condition was the fuel-saving condition. Table 2 illustrated the details of the test results. As the diesel engine load increased, the total PMs emission showed an increasing trend. However, upstream PM emissions of unit power decreased from 1.1148g/kW·h to 0.432g/kW·h. It might be because the low load operation caused the incomplete combustion of fuel and led to a higher PM emissions phenomenon. And with the increase in diesel engine load, the

**Table 2. Upstream and downstream PMs emissions of unit power (g/kW-h) and PMs filtering efficiency (%) of DPF at different diesel engine loads.**

| No. | Load | PMs emissions of unit power (g/kW-h) | | Filtering efficiency (%) |
|---|---|---|---|---|
| | | Upstream | downstream | |
| 1 | 25% | 1.1148 | 0.2428 | 78.2% |
| 2 | 50% | 0.7637 | 0.1878 | 75.4% |
| 3 | 65% | 0.6417 | 0.1713 | 73.3% |
| 4 | 75% | 0.5683 | 0.1866 | 67.2% |
| 5 | 100% | 0.4320 | 0.2765 | 36.0% |

upstream PM emissions decreased gradually. Meanwhile, the downstream PMs emissions were less than upstream ones, indicating that the DPF module had a positive impact on PMs filtering.

But as the diesel engine load increases, the PMs filtering efficiency showed a downward trend, especially after 75% load. The phenomenon indicated that the DPF module was likely to store some sulfur contents in the exhaust gas at low loads operation. When the engine load and the exhaust gas temperature increased, those stored sulfur contents would react with other components in the exhaust gas to provide more nucleated cores for volatile organic compounds (VOC). As a result, that increased downstream small-size nucleation mode PMs. Although several studies pointed out that nucleation mode PMs, mainly composed of soluble organic fraction (SOF), could be easily captured and oxidized by DPF due to Brownian Motion, PMs were likely to penetrate through the DPF due to high space velocity, beyond the designed one. [26] And that resulted in a drop in PMs filtering efficiency.

We conducted smoke tests with the smoke meter for 65%, 75%, and 100% load for further investigation. Table 3 illustrated the details of the test results. Compared to partial stream dilution sampling and filter paper weighing methods, the smoke meter was more sensitive to large-size accumulation and coarse mode PMs and less sensitive to nucleation mode and some non-absorbent macromolecules PMs [27]. As can be seen from Table 3, although PMs filtering efficiency showed a decreasing trend as load increased, it still maintained above 90%, much higher than that measured by filter paper weighing methods. It was noteworthy that upstream PMs emission of DPF module showed a peak at 75% load, followed by a decrease at 100% load. That was primarily because 75% load was the designed load of the fuel-saving mode for the diesel engine, and the lower fuel gas temperature caused by the fuel-saving model led to a rise in PMs. Considering that, the conclusion could be that the change in upstream and downstream smoke levels of DPF was mainly dependent on diesel engine operation load. The more load the diesel engine operated at, the higher the smoke level was.

Although some scholars pointed out that PMs would change from nucleation mode to accumulation mode as the diesel engine load increased [28,29], the DPF module temperature would also go up with the accumulation of PMs. And that would drive accumulation mode PMs to decompose into nucleation mode. As a result, there was no obvious smoke level increase with the rise of engine load and nucleation mode PMs would escape the DPF module

**Table 3. Upstream and downstream smoke level (mg/m$^3$) and filtering efficiency (%) of DPF module under different loads.**

| No. | Load | Smoke level (g/kW-h) | | Filtering efficiency (%) |
|---|---|---|---|---|
| | | Upstream | Downstream | |
| 1 | 65% | 53.11 | 0.17 | 99.7% |
| 2 | 75% | 59.33 | 0.23 | 99.6% |
| 3 | 100% | 56.66 | 3.29 | 94.2% |

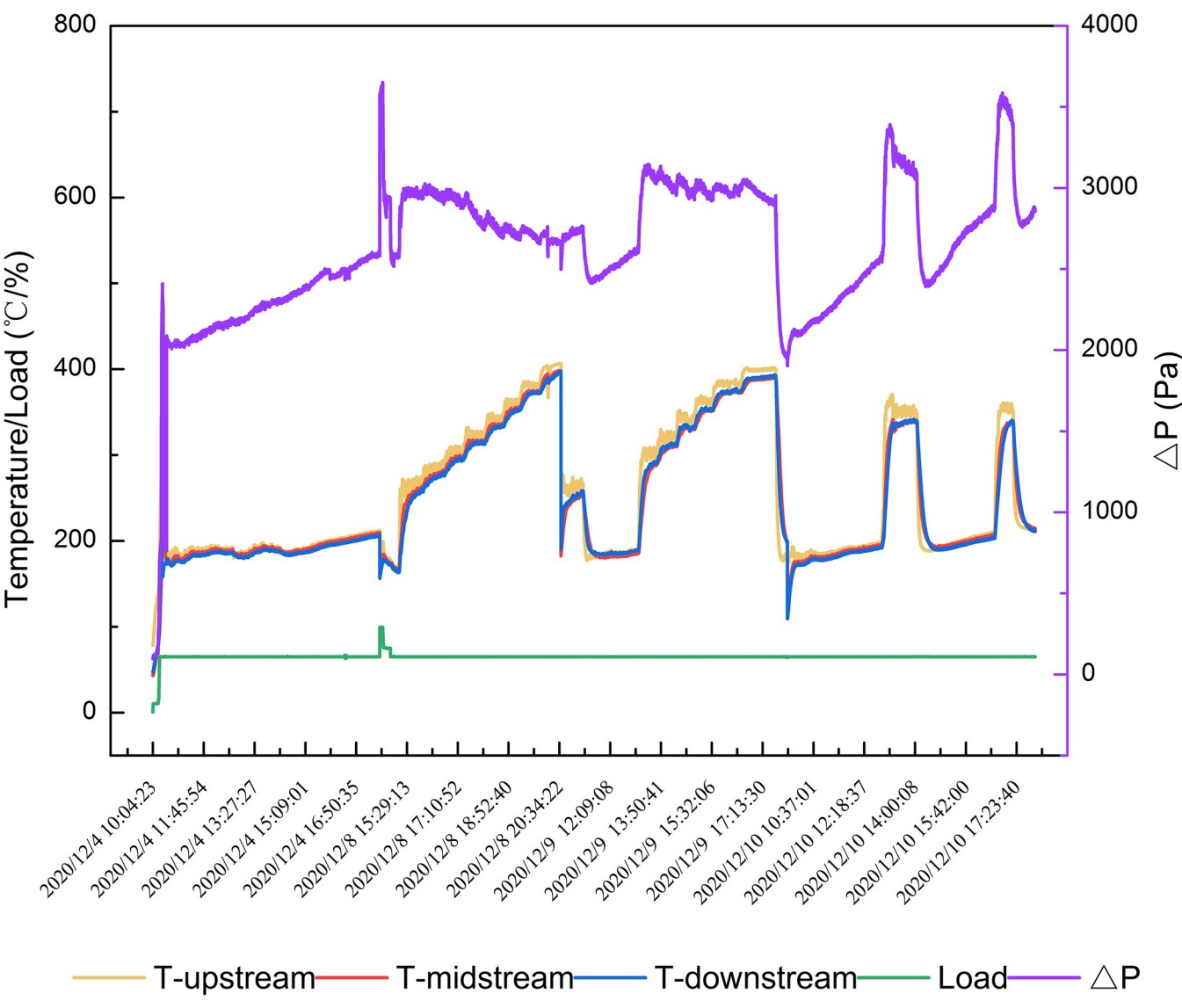

**Fig 6. DPF BPT tests in pilot-scale bench.**

at high airspeeds, resulting in a significant decrease in PMs emissions of unit power at high load operations [30,31].

**DPF module regeneration durability.** As mentioned in Section 3.1, BPT was one of the most vital parameters for judging DPF module performance. But due to the differences between the bench-scale engine and the pilot-scale one, we took another test on DPF BPT on pilot scale engines. At the beginning of the tests, the diesel engine load was 65%, and the initial exhaust gas temperature was 200˚C by the operation of the water-cooling system. The test lasted for four cycles. Fig 6 showed and the test results.

When the exhaust gas temperature was 300˚C, the pressure drop started to decrease obviously, indicating that the catalytic oxidation rate of PMs was over the deposition rate at 300˚C. With the exhaust gas temperature exceeding 300˚C, the pressure drop kept decreasing while the DPF module went into a regeneration state.

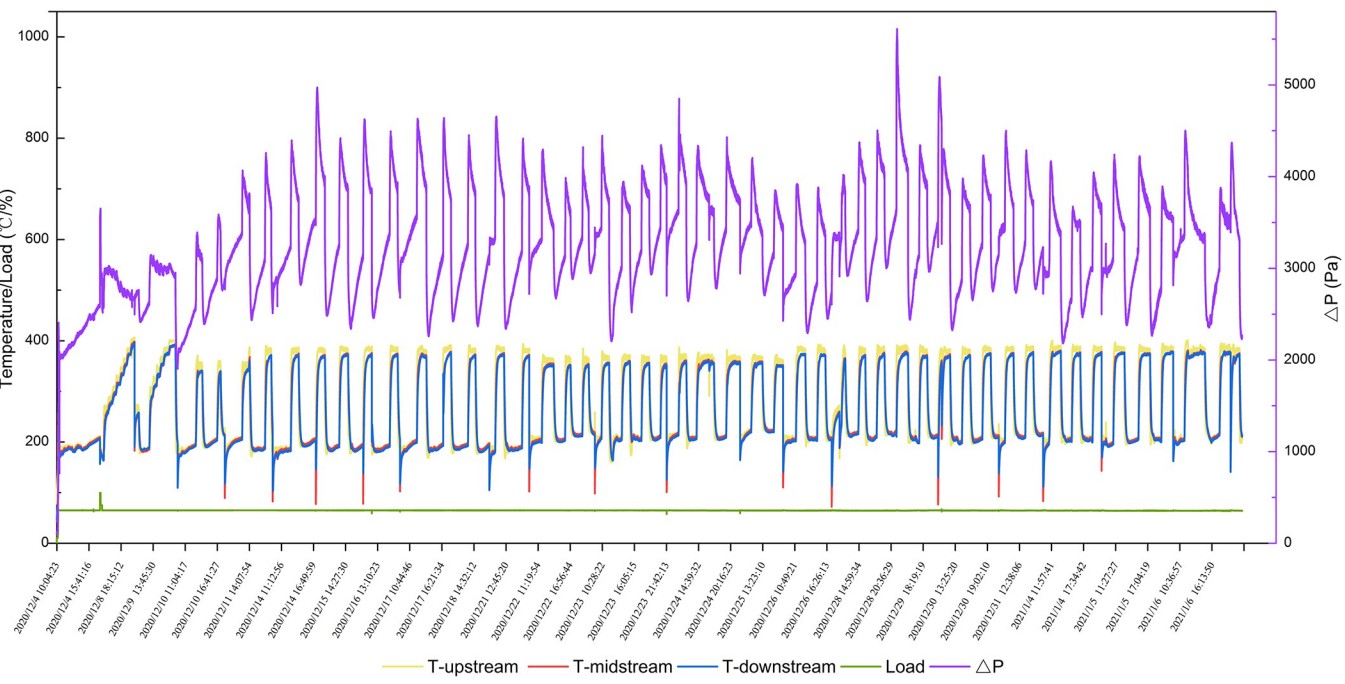

**Fig 7. Long-term DPF module regeneration durability test.**

After testing BPT, we tested the regeneration durability of the DPF module for 48 cycles lasting 205h in total. The initial engine load was 65%, and the exhaust gas temperature was 200°C. As time went on, the PMs deposited on the DPF module. With more PMs depositing on the DPF module, the exhaust gas temperature increased, and the DPF module went into a regeneration state. As shown in Fig 7, the DPF module had excellent regeneration and durability, since the pressure drop could go back to its initial value after PMs deposition and oxidation. The DPF regeneration lasted for 1.5-2h at 380°C.

We disassembled and checked the DPF module at the end of the tests, as shown in Fig 8. As operating cycles increased, some PMs were observed upstream and inside of the DPF module. But the DPF catalyst was structurally sound, with no cracks or breakage present, indicating DPF was with excellent regeneration and durability.

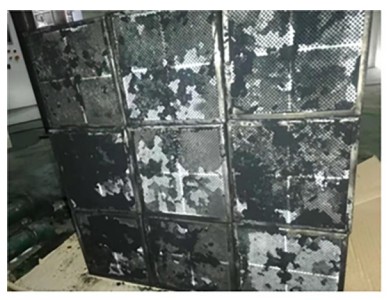
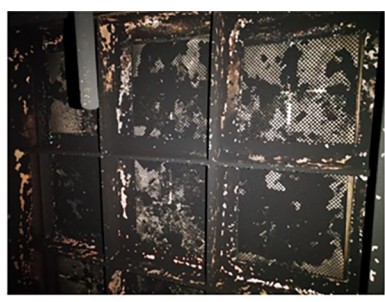
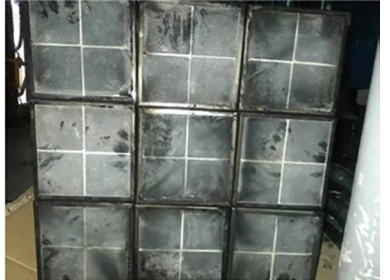

(A)  (B)  (C)

**Fig 8. DPF module.** (A) Upstream; (B) Inside; (C) Downstream.

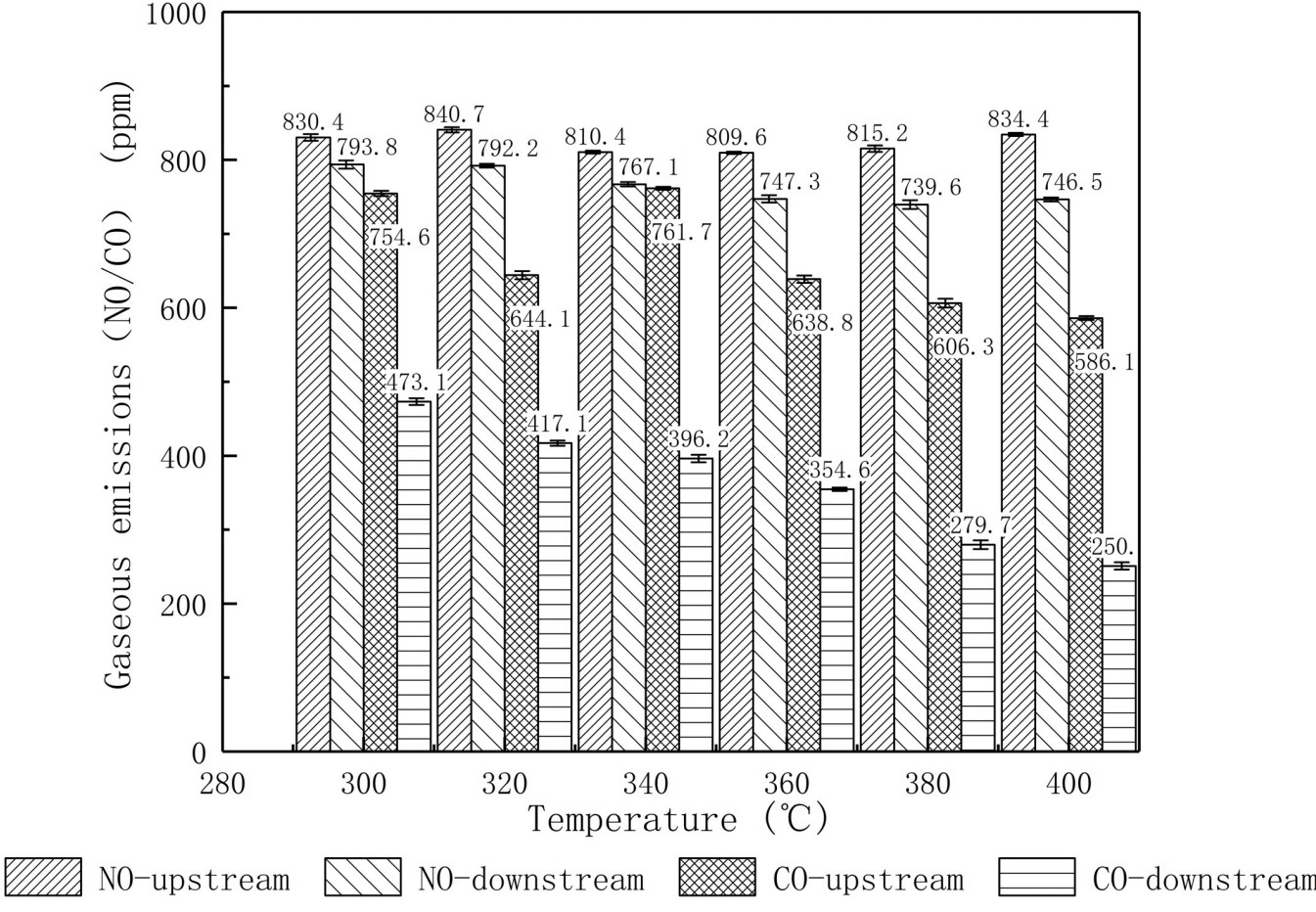

**Fig 9. Upstream and downstream gaseous emissions of DPF.**

**Gaseous emissions control.** Gaseous emissions control was another factor evaluating DPF performance, so we measured upstream and downstream gaseous emissions of DPF at different exhaust gas temperatures. Considering cost optimality and DPF BPT, we conducted the test at 65% load of diesel engines.

$NO_x$ emissions are mainly caused by the oxidation of $N_2$ in the air for diesel engine cases, and almost 90% of which is NO. There is few Fuel $NO_x$ and Prompt $NO_x$. And CO emissions are mainly from incomplete combustion of fuel. As mentioned in Section Introduction, PM removal is realized by oxidization of $NO_2$, which is transformed from NO by precious metals catalyzation. As a result, NO will be also oxidized during the PM removal process. Besides, CO will also be oxidized to $CO_2$ due to the strong oxidizing of precious metals. As shown in Fig 9, the test results told that downstream NO and CO emissions of DPF showed a decreasing trend as the exhaust gas temperature increased. The conversion rates of NO and CO were 10.53% and 57.19%, respectively, at 360˚C.

Since the active sites in the precious metal Pt could effectively increase CO conversion, the DPF module was more effective for CO conversion than NO. It had been noted that changes in the Pt state and the interaction between the precious metals would affect the conversion efficiency of NO [32], so there was an optimal ratio of Pt for the NO conversion. That might explain DPF module had a better conversion performance of CO than NO.

## Conclusions

Sulfur-resisting material can effectively limit DPF inactivation for those low-speed marine engines fueled with HFO. This paper investigated the performance of DPF loaded over with sulfur resisting material containing ($Al_2O_3$, $BaSO_4$, and organic colloid) and Pt(40g/cft), and the results told that:

1. The BPT of the DPF module decreased from 325˚C to 300˚C after sulfur-resisting and Pt loading treatment. And that also improved sulfur resisting performance. The PM residuals and accumulated PM residuals decreased by 0.6% and 14%, respectively.

2. As diesel engine load increased from 10% to 100%, total PM emissions increased, and PM emissions of unit power decreased from 82.28% to 36%. But DPF module still had a good performance in accumulation mode PMs control, above 90%.

3. The DPF module was excellent in regeneration performance, with BPT being 300˚C. The 205h endurance test showed that the DPF catalyst also had good regeneration durability. The pressure drop could go back to the initial value after regeneration, with a regeneration period of 1.5-2h at 380˚C. As the operating cycles increased, accumulated PMs were observed upstream and inside the DPF module, but the DPF module structure remained good with no cracks or breakage.

4. The DPF module had the effect of gaseous emission control, and the conversion rate of NO and CO was 10.53% and 57.19%, respectively.

For later studies, Pt content and recipe optimization of sulfur resisting material might further improve DPF performance for those low-speed marine engines fueled with HFO.

## Supporting information

**S1 File. Contains all the supporting tables.**
(PDF)

**S1 Data.**
(PDF)

## Acknowledgments

This research was supported by Shanghai Marine Diesel Engine Research Institute.

## Author Contributions

**Conceptualization:** Xiaobo Li, Diming Lou.

**Data curation:** Yaqiong Liu.

**Formal analysis:** Shien Tu.

**Methodology:** Ke Li.

**Project administration:** Teng Shen.

**Resources:** Zhigang Wang.

**Validation:** Haoran Yang, Zhigang Wang.

**Writing – original draft:** Diming Lou.

**Writing – review & editing:** Haoran Yang, Diming Lou.

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
