## [Decision Letter · Decision Letter 0]

13 Jun 2022

PONE-D-22-13038Experimental evaluation of DPF performance loaded over Pt and sulfur resisting material for marine diesel enginesPLOS ONE

Dear Dr. Yang,

Thank you for submitting your manuscript to PLOS ONE. After careful consideration, we feel that it has merit but does not fully meet PLOS ONE’s publication criteria as it currently stands. Therefore, we invite you to submit a revised version of the manuscript that addresses the points raised during the review process.

We look forward to receiving your revised manuscript.

Kind regards,

Xiaowei Zhang, Ph.D.

Academic Editor

PLOS ONE

Journal Requirements:

2. PLOS requires an ORCID iD for the corresponding author in Editorial Manager on papers submitted after December 6th, 2016. Please ensure that you have an ORCID iD and that it is validated in Editorial Manager. To do this, go to ‘Update my Information’ (in the upper left-hand corner of the main menu), and click on the Fetch/Validate link next to the ORCID field. This will take you to the ORCID site and allow you to create a new iD or authenticate a pre-existing iD in Editorial Manager. Please see the following video for instructions on linking an ORCID iD to your Editorial Manager account: https://www.youtube.com/watch?v=_xcclfuvtxQ.

3. Please include a caption for figure 9.

Reviewers' comments:

Reviewer's Responses to Questions

**Comments to the Author**

1. Is the manuscript technically sound, and do the data support the conclusions?

Reviewer #1: Yes

Reviewer #2: Yes

Reviewer #3: Yes

2. Has the statistical analysis been performed appropriately and rigorously? 

Reviewer #1: Yes

Reviewer #2: Yes

Reviewer #3: Yes

3. Have the authors made all data underlying the findings in their manuscript fully available?

Reviewer #1: Yes

Reviewer #2: Yes

Reviewer #3: Yes

4. Is the manuscript presented in an intelligible fashion and written in standard English?

Reviewer #1: Yes

Reviewer #2: Yes

Reviewer #3: Yes

5. Review Comments to the Author

Reviewer #1: Different from vehicle engines, DPF inactivation is an unavoidable issue for marine diesel engines fueled with High Sulfur Fuel Oil. This paper introduced a sulfur resisting material in SiC-DPF to improve DPF module performance. The results of bench-scale experiments showed that the Balance Point Temperature of the modified DPF module was 300°C and DPF modules had a good filtration performance, PMs residual being less than 0.6 g per cycle. In pilot-scale tests, PMs emissions of unit power decreased with engine load going up, filtration efficiency of nucleation mode PMs being only 36% under 100% load, while DPF still had a good performance in accumulation mode PMs control, being 94.2% under the same load. DPF modules showed excellent regeneration durability in the 205h endurance test, with a regeneration period of 1.5-2h under 380°C. There was no obvious degeneration in the DPF module structure, with no cracks or breakage. Besides, the DPF module could also control gaseous emissions, total emissions decreased by 10.53% for NO and 57.19% for CO, respectively.

I think this is an important and interesting study work on the investigating DPF performance for marine diesel engines fueled with High Sulfur Fuel Oil, even if the current regulation do not limit the PM emissions in IMO, so I recommend to publish this paper on the journal after some necessary revisions.

Firstly, the High Sulfur Fuel Oil only was used for low-speed marine engines, but that engines should according to the IMO Tier 3, while the STAGE II, effective since July 2021 should be near area of seaside, so the author should clarify that.

Secondly, the literature reviews on recently relevant works on energy and emissions of marine engines should be added into Introduction, such as Fuel, 139: 472-481, 2015; Combustion and Flame 227 (2021): 52-64; Fuel 310 (2022) 122307, to make the literature more completed and help to analyze the current study results.

Finally, the format of the manuscript should be revised according to the guideline of the Journal, such as some of titles keep the first characteristic capital, but some of them are not, which should keep the same.

Reviewer #2: All abbreviations must be given full where first in use.

Some specific results should be given in the abstract.

What is the main aim of this study? It should be specified at the end of the introduction.

NOx emissions occurring mechanism should be defined and what mechanism did affect this study?

What is the main result of this study? It should be indicated at the end of the conclusion section and a few recommendations must be given for further studies.

Reviewer #3: 1. Why the load is not taken as 0, 25, 50, 75 & 100?

2. Any uncertainty analysis?

3. What fuel is used? Is this methodology applicable to other alternate fuels?

4. How you do regeneration?

5. What is recycle time?

6. How you do oxidation to remove PM?

7. Any engine specifications?

8. Gas and Smoke analyzer specifications?

9. What is sulphur resistant material?

10. What is figure 4a,4b,4c represents?

6. PLOS authors have the option to publish the peer review history of their article (what does this mean?). If published, this will include your full peer review and any attached files.

Reviewer #1: No

Reviewer #2: No

Reviewer #3: **Yes: **Balaji Gnanasikamani

---

## [Author Response · Author response to Decision Letter 0]

17 Jul 2022

Journal Requirements:

https://journals.plos.org/plosone/s/file?id=wjVg/PLOSOne_formatting_sample_main_body.pdf and https://journals.plos.org/plosone/s/file?id=ba62/PLOSOne_formatting_ sample_title_authors_affiliations.pdf

Respond: We are sorry for neglecting the PLOS ONE's style requirements, we have revised our manuscript according to the PLOS ONE style templates.

2. PLOS requires an ORCID iD for the corresponding author in Editorial Manager on papers submitted after December 6th, 2016. Please ensure that you have an ORCID iD and that it is validated in Editorial Manager. To do this, go to ‘Update my Information’ (in the upper left-hand corner of the main menu), and click on the Fetch/Validate link next to the ORCID field. This will take you to the ORCID site and allow you to create a new iD or authenticate a pre-existing iD in Editorial Manager. Please see the following video for instructions on linking an ORCID iD to your Editorial Manager account: https://www.youtube.com/watch?v=_xcclfuvtxQ.

Respond: We have updated the ORCID iD information.

3. Please include a caption for figure 9.

Respond: We have included a caption for figure 9.

Thank you for your valuable comment.

Reviewer #1:

1.Firstly, the High Sulfur Fuel Oil only was used for low-speed marine engines, but that engines should according to the IMO Tier 3, while the STAGE II, effective since July 2021 should be near area of seaside, so the author should clarify that.

Respond: We have revised the obscure description in our manuscript. 

The revised description is as follows:

Although there is no global regulation, many countries and organizations are drafting local regulations on limiting marine PM emissions. International Maritime Organization (IMO) is introducing Regulation 14 inside Emission Control Areas (ECA) to limit SOx and PMs emission. China has carried out “Limits and measurement methods for exhaust pollutants from marine engines (STAGE II, effective since July 2021)” for emission control in the coastal area. [5-7]

2.Secondly, the literature reviews on recently relevant works on energy and emissions of marine engines should be added into Introduction, such as Fuel, 139: 472-481, 2015; Combustion and Flame 227 (2021): 52-64; Fuel 310 (2022) 122307, to make the literature more completed and help to analyze the current study results.

Respond: Thanks for providing valuable papers to help to support the current study results. We have cited these paper in our manuscript.

3.Finally, the format of the manuscript should be revised according to the guideline of the Journal, such as some of titles keep the first characteristic capital, but some of them are not, which should keep the same.

Respond: We are sorry for neglecting the PLOS ONE's style requirements, we have revised our manuscript according to the PLOS ONE style templates. 

Thank you for your valuable comment.

Reviewer #2:

1.All abbreviations must be given full where first in use.

Respond: We revised all abbreviations with full terms where first in use.

2.Some specific results should be given in the abstract.

Respond: We revised the abstract with some specific results.

3.What is the main aim of this study? It should be specified at the end of the introduction.

Respond: Although DPF has been confirmed as a feasible way of PM filtering, there is few successful DPF application case in the low-speed marine diesel engines field, due to the S content in HFO. As a result, we introduced a sulfur resisting material to improve the DPF performance for low-speed marine diesel engines. We believe the findings in our paper can provide a theoretical and technical basis for the practical application of DPF loaded over sulfur resisting material. This is the main aim of this study. We have specified it at the end of the introduction as reviewer recommended.

4.NOx emissions occurring mechanism should be defined and what mechanism did affect this study?

Respond: NOx emissions are mainly caused by the oxidation of N2 in the air for diesel engine cases, and almost 90% of which is NO. There is few Fuel NOx and Prompt NOx. And CO emissions are mainly from incomplete combustion of fuel. As mentioned in Section Introduction, PM removal is realized by oxidization of NO2, which is transformed from NO by precious metals catalyzation. As a result, NO will be also oxidized during the PM removal process. Besides, CO will also be oxidized to CO2 due to the strong oxidizing of precious metals.

We revised this part in Section Gaseous emissions control.

5.What is the main result of this study? It should be indicated at the end of the conclusion section and a few recommendations must be given for further studies.

Respond: We revised the Section Conclusion with main results and further study recommendations.

The revised description is as follows:

Sulfur-resisting material can effectively limit DPF inactivation for those low-speed marine engines fueled with HFO. This paper investigated the performance of DPF loaded over with sulfur resisting material containing (Al2O3, BaSO4, and organic colloid) and Pt(40g/cft), and the results told that:……

AND

For later studies, Pt content and recipe optimization of sulfur resisting material might further improve DPF performance for those low-speed marine engines fueled with HFO.

Thank you for your valuable comment.

Reviewer #3:

1. Why the load is not taken as 0, 25, 50, 75 & 100?

Respond: The initial test plan only contained 25, 50, 75 & 100 load initially. During tests, we had 10% and 65% included. For 10%, we planned to explore the extreme working condition of DPF and engines. And for 65%, we found the exhaust gas temperature under 65% load was close to DPF balance point temperature. That’s why we took 10% and 65% load in tests. After we read Reviewer’s comments, we also thought that 10% load condition had no obvious contribution in our manuscript and we deleted that part from our manuscript. But we still kept 65% load condition in our manuscript to reduce the influence that 75% load condition (fuel-saving condition) might have on analysis. And we made an explanation for that in Section DPF module PMs filtering efficiency.

2. Any uncertainty analysis?

Respond: We are sorry for the missing of uncertainty analysis. We have revised our manuscript with uncertainty analysis.

3. What fuel is used? Is this methodology applicable to other alternate fuels?

Respond: The fuel used is an artificial high-sulfur diesel fuel, which was made by adding thiophene into National VI diesel fuel (GB/19147-2016), and the total sulfur content of diesel fuel is 0.5%. According to https://10.1016/j.cjche.2016.08.030 and https://doi.org/10.1155/2021/4762184, these two papers also take this methodology to varying the S content in diesel fuel in investigation. As a result, we believe this methodology is applicable to other alternate fuels.

4. How you do regeneration?

Respond: DPF Regeneration usually gets realized via the heat of the exhaust gas (passive regeneration) or extra fuel injected into the device (active regeneration). In our paper, we conduct it through passive regeneration. And we have revised our manuscript with a clear description about regeneration in Section DPF module BPT experiments.

The revised part is as following:

As the DPF module filtered PMs, the engine exhaust backpressure, exhaust gas temperatures, and DPF module temperature would rise. Then, the trapped PMs would become oxidized with the environment temperature increase, which led to a decrease in engine exhaust backpressure.

5. What is recycle time?

Respond: We are sorry for these misleading expressions. The meaning we want to express is the number of cycles that DPF have worked. We have revised all these expressions in figures in our manuscript.

6. How you do oxidation to remove PM?

Respond: We introduced precious metals in DPF to realize low-temperature oxidation of PMs. During DPF working process, precious metals help to catalyze NO to NO2 under low temperature, and NO2 will further oxidize PM particulates to CO2. 

We have revised our manuscript with a clear description about how the DPF removes PM in Section Introduction.

7. Any engine specifications?

Respond: The engine used in simulated-exhaust-gas experiments is BJ493ZLQ4 from Beijing Futian Environmental Power Co., Ltd. It is a four-stroke, high-pressure common rail system, turbocharged and intercooled engine. The displacement is 2.8L, the rated power is 70 kW, and the declared speed is 3600rpm.

The engine used in pilot-scale tests was WHM6160C550-5 from Weichai Holding Group Co., Ltd. It was a four-stroke, water-cooled, in-line, turbocharged, and intercooled engine. The displacement was 24.12L, the rated power was 405 kW, and the declared speed was 1500rpm.

We have revised our manuscript with some engine specifications in Section Experiments and tests.

8. Gas and Smoke analyzer specifications?

Respond: CEMS used was the AVL AMA i60 and smoke meter used was the Testo 338smoke tester. We have revised this description in Section Experiments and tests.

9. What is sulphur resistant material?

Respond: The sulfate resisting material is named as SR-1. It is a mixture of multiple industrial-batch raw materials, containing Al2O3, BaSO4, and organic colloid.

We have make it clear in Section Sample preparation.

10. What is figure 4a,4b,4c represents?

Respond: Fig 4. presents the BPT results of DPF modules with different coating materials and Pt loadings in simulated-exhaust-gas bench-scale experiments. We compare there DPF modules performance and the results will help us to select the most suitable DPF modules for pilot-scale tests.

Thank you for your valuable comment.

---

## [Editor Report · Decision Letter 1]

20 Jul 2022

Experimental evaluation of DPF performance loaded over Pt and sulfur resisting material for marine diesel engines

PONE-D-22-13038R1

Dear Dr. Yang,

We’re pleased to inform you that your manuscript has been judged scientifically suitable for publication and will be formally accepted for publication once it meets all outstanding technical requirements.

Kind regards,

Xiaowei Zhang, Ph.D.

Academic Editor

PLOS ONE
---

## [Editor Report · Acceptance letter]

29 Jul 2022

PONE-D-22-13038R1 

Experimental evaluation of DPF performance loaded over Pt and sulfur-resisting material for marine diesel engines 

Dear Dr. Yang:

I'm pleased to inform you that your manuscript has been deemed suitable for publication in PLOS ONE. Congratulations! Your manuscript is now with our production department. 

Kind regards, 

on behalf of

Dr. Xiaowei Zhang 

Academic Editor

PLOS ONE